# Towards Understanding Deep Policy Gradients:
# A Case Study on PPO

**Buhua Liu**
Department of Computer Science
Hong Kong Baptist University
Kowlong Tong, Hong Kong
`csbhliu@comp.hkbu.edu.hk`

**Chong Yin**
Department of Computer Science
Hong Kong Baptist University
Kowlong Tong, Hong Kong
`chongyin@comp.hkbu.edu.hk`

## Abstract

Deep reinforcement learning has shown impressive performance on many decision-making problems, where deep policy gradient algorithms prevail in continuous action space tasks. Although many algorithm-level improvements on policy gradient algorithms have been proposed, recent studies have found that code-level optimizations also play a critical role in the claimed enhancement. In this paper, we further investigate several code-level optimizations for the popular Proximal Policy Optimization (PPO) algorithm, aiming to provide insights into the importance of different components in the practical implementations.[1]

## 1 Introduction

In the reinforcement learning (RL) setting, an agent can solve complex decision-making problems by trial and error, specifically, the agent can improve its policy by interacting with a stateful environment with the goal of maximizing cumulative reward. Leveraging the power of deep learning, deep reinforcement learning algorithms have achieved great success even in many challenging games, such as Honor of Kings [6]. However, despite such impressive results, deep RL algorithms are still not as reliable as the (deep) supervised learning counterparts and they are shown to be brittle and hard to reproduce and even unreliable across runs [1].

Motivated by the reproducibility issues facing deep RL, we review the literature and investigate various components in the practical implementations of deep policy gradient algorithms. In particular, Engstrom et al. [1] find that besides the algorithm-level optimization introduced in deep policy gradient methods, e.g., from Trust Region Policy Optimization (TRPO) [4] to Proximal Policy optimization (PPO) [5], the code-level optimizations also play a key role in the claimed performance improvements, such as value function clipping, reward scaling, initialization, layer scaling, adam learning rate annealing, reward clipping, observation normalization, observation clipping and so on [1]. Apart from the literature review, during our implementation of the course assignments, we also observed that the initialization of policy function in HW1 as well as the parameter initialization in HW3 largely impact the final performance of our trained policy in terms of the mean episode reward, which indicates that even small modifications to the core algorithm may also have a large impact on the final model performance.

Based on the work of Engstrom et al. [1], we further investigate several code-level optimizations towards better understanding of the importance of different components in modern deep reinforcement learning algorithms. Specifically, we conduct ablation studies in the MuJoCo environment on four code-level optimizations including rewarding clipping, observation clipping, global gradient clipping, and observation normalization, aiming to investigate the contribution of each component to the best model performance in terms of maximum reward achieved.

---

[1] Video presentation is available at `https://youtu.be/MOuTLoEUwGQ`

## 2 Motivating Papers

Ilyas et al. [3] studied how the behavior of deep policy gradient algorithms reflects the conceptual framework motivating their development. Specifically, they identified three key elements of current framework of deep policy gradient algorithms, i.e., gradient estimation, value prediction, and optimization landscapes. From the three perspectives, they conducted comprehensive experiments which reveal several unexpected observations that are not explained by the current theory. In other words, they found that there is a significant gap between the theory motivating current algorithms and the actual mechanisms improving the performance. Engstrom et al. [1] conducted a thorough evaluation of the code-level optimizations via a case study on TRPO and PPO, which are typically considered as implementation details and therefore are often understated in papers, surprisingly, these elements are shown to have major implications for the performance of both algorithms. Henderson et al. [2] investigate the key factors affecting reproducibility including non-determinism in standard benchmark environments and variance intrinsic to the algorithms.

## 3 Problem Formulation

Inspired by the findings of [3, 1, 2], we aim to explore the implications of different components in practical implementations of popular deep policy gradient algorithms. Engstrom et al.[1] point out 9 code-level optimizations, which may appear as insignificant changes to the core algorithm optimization method, apart from the already reported ones, we study the other four code-level optimizations, specifically, reward clipping, global gradient clipping and observation normalization, which we illustrate in details below.

### 3.1 Reward Clipping

Each game would have different score scales which would make training unstable. The reward clipping technique will clip the scores into an predefined interval.

$$R_t(\tau) = \sum_{t'=t}^{\infty} \gamma^{t'-t} r_{t'} \tag{1}$$

$$R_t = \begin{cases} -\delta, & R_t < -\delta_1 \\ \delta, & R_t > \delta_1 \\ R_t, & otherwise \end{cases} \tag{2}$$

where $\delta_1$ indicates the predefined threshold for the reward value.

### 3.2 Observation Clipping

The observation $S_t$ describes the state of the environment.

$$S_t = \begin{cases} -\delta, & S_t < -\delta_2 \\ \delta, & S_t > \delta_2 \\ S_t, & otherwise \end{cases} \tag{3}$$

where $\delta_2$ indicates the predefined threshold for the state representations.

### 3.3 Global Gradient Clipping

The gradient clipping is a popular strategy to make the gradient bounded to avoid unstable training.

$$g = min\{1, \frac{\delta_3}{||g||}\}g \tag{4}$$

where $g$ represents the gradient, $\delta_3$ represents the gradient norm threshold.

### 3.4 Observation Normalization

$$S = \frac{S - mean(S)}{Var(S)} \tag{5}$$

# 4 Experiments

## 4.1 Initial Attempts

Gradient normalization is a common strategy to improve the learning stability and model performance. We first conduct ablation study on different norms for gradient normalization. Specifically, we compare the performance of agents trained with five different norms in the CartPole-v0 environment. As shown in Figure Figure 1, the smallest singular and direct gradient normalization have a poorer performance compared with other three ones. The other three are more stable and the nuclear normalization can lead a faster convergence.

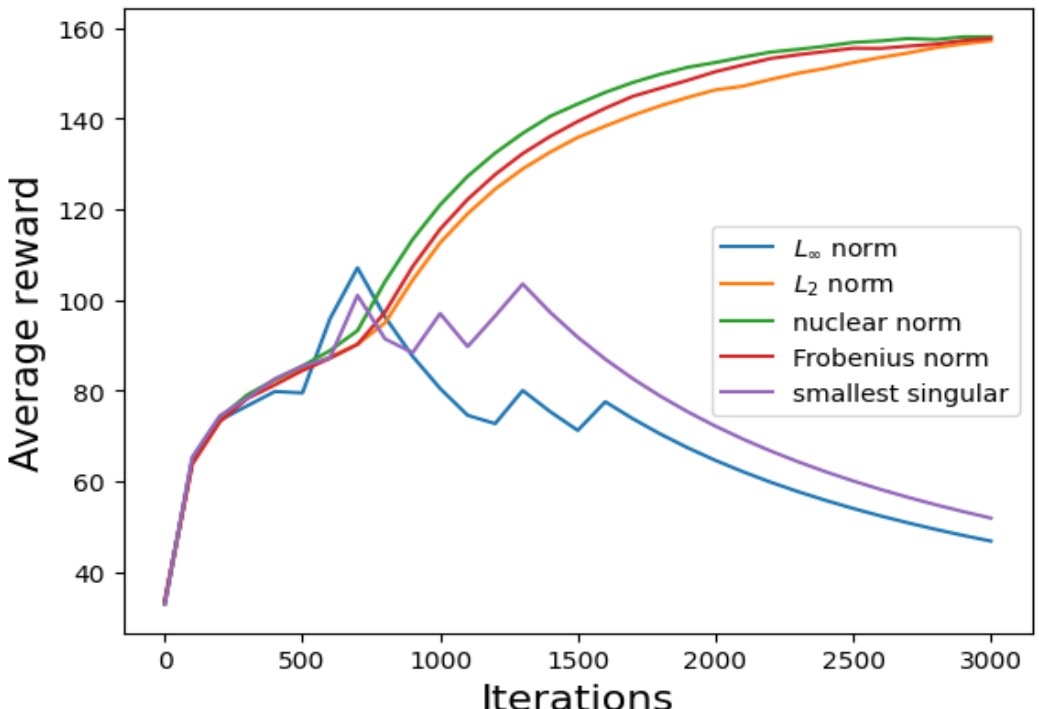

Figure 1: Average reward as a function of training iterations trained with MLP as the Q-value apporoximator in CartPole-v0 environment. Here we compare the performance of five gradient clipping strategies: direct gradient clipping ($L_\infty$ norm), largest singular normalization ($L_2$ norm), nuclear normalization (nuclear norm), Frobenius norm (Frobenius norm), and smallest singular value normalization (smallest singular).

## 4.2 Main Experiments

### 4.2.1 Setup

In this section, we conduct an ablation study on aforementioned four code-level optimizations with PPO algorithm in the Hopper-v2 environment.

| | |
|---|---|
| Timesteps per iteration | 2048 |
| Discount factor $\gamma$ | 0.99 |
| GAE discount $\lambda$ | 0.95 |
| Value network learning rate | 1.5e-4 |
| Value network num. epochs | 10 |
| Entropy coeff. | 0 |

Table 1: Hyper-parameter settings for PPO in Hopper-v2

The ablation settings are as follows:

- **Reward Clipping** The rewards are clipped to be less than 5.0 or 10.0.
- **Observation Clipping** The observations are clipped with $l_2$ norm less than 5.0 or 10.0.
- **Global Gradient Clipping** The gradients are clipped with $l_2$ norm less than 0.5 or no clipping.
- **Observation Normalization** The states are normalized with their mean and variance or no normalization.

We take policy learning rate (Adam) with four values (1e-5, 8e-5, 1.5e-4, 2.2e-4) and five trails to form our full experiments (in total $4 \times 5 \times 2^4 = 320$ trials).

### 4.2.2 Results

Figure 2 shows the histogram of the average rewards of the agents trained with four configurations. The results show that each optimizations have varying degrees of impact on the final rewards, in particular, the reward clipping and state normalization have a significantly larger impact on the average rewards.

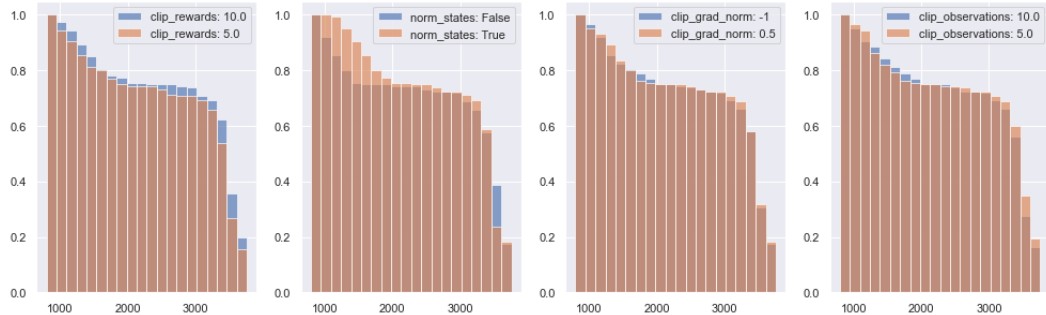

Figure 2: An ablation study on the four optimizations. From the left to right, the figure represent reward clipping, observation clipping, global gradient clipping and observation normalization, respectively. For each of $2^4$ possible configurations of the optimizations, we train the agent in Hopper-v2 using PPO. Different colors indicate different configuration of code-level optimizations.

## 5   Conclusion and Future Work

In this paper, we investigate the impact of several code-level optimizations to the model performance in terms of mean reward achieved through extending the framework of Engstrom et al. [1]. Our results demonstrate that the deep policy gradient algorithms, in particular, PPO, exhibit high sensitivity to various subtle code modifications. To better understand the interaction between various deep policy gradient algorithms and environments, we leave for future work the investigation to how different algorithms behave in different environments.

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
