# OpenReview forum: "Towards Understanding Deep Policy Gradients: A Case Study on PPO"
_CUHK.edu.hk/2021/Course/IERG5350_

### Official Review · AnonReviewer1 · 2020-12-15
**This paper tested the impact of different parameters and the use of different variable normalization on PPO performance in the Hopper-v2 environment. This paper is not innovative, but it deepens the students' understanding of the basic algorithm.**

**Rating:** 6
**Confidence:** 5

**Review:**

General:

1. Significance: The main contribution of this paper is to test the impact of different parameters and the use of different variable normalization on PPO. Nothing new. But it may help students understand the PPO and basic settings.
2. Novelty: Nothing.
3. Technical quality: The parameter adjustment, normalization , and clip designed in this paper are not technical.
4. Clarity: The structure of this paper is clear, but it lacks a flowchart that reflects the overall method. The analysis of the results is not detailed. The author only explains that these settings have an impact, but how? Will the result be better or worse? Under what cases should I use what settings to improve performance? These more important conclusions were not obtained.

Specific:

1. Pros:
a. The PPO algorithm is implemented and the influence of the parameters is tested.
b. Tested the influence of basic operations such as normalization and clip.
2. Cons:
a. Without any innovation.
b. No practical meaning.
c. The content is too simple.

---

### Official Review · AnonReviewer3 · 2020-12-20
**The main contribution of this paper is to  investigate the impact of several code-level optimizations to the PPO model performance. But no novelty and lack of detailed experiments。**

**Rating:** 4
**Confidence:** 5

**Review:**

Significance: The main contribution of this paper is to  investigate the impact of several code-level optimizations to the PPO model performance.
Novelty: No.
Technical quality: Not very well.
Clarity: The paper investigate the impact of several optimizations. But lack of detailed experimental details and effect analysis and no experiment to support the conclusion.
Specific:
Pros: Seveal optimizations are tried such as normalization and clip.
Cons: a. Without any innovation. b. No practical meaning. c. The content is too simple. d. The experiment does not enough.

---

### Official Review · AnonReviewer2 · 2020-12-20
**A work that evaluated the performance of PPO on some classical RL environments by varying the algorithm implementation details.**

**Rating:** 5
**Confidence:** 5

**Review:**

**General:**
The paper tried to address the problem that reinforcement algorithms' performance is usually not stable and can not be easily reproduced. Therefore, they evaluated the performance of PPO with different code implementations on  CartPole-v0 environment and Hopper-v2 environment.

**Evaluation of the quality:**
The work is kind of like an ablation study on PPO. Experiments showed the performance of PPO can be affected by the choice of gradient normalization, reward clipping, observation clipping, global gradient clipping, and observation normalization.

**Clarity:**
The work is clearly written.

**Originality:**
There is no much originality in the investigated problem.

**Significance:**
The investigated problem is kind of trivial. Observing from the figures shown in the paper, varying some settings doesn't seem to affect the performance of PPO that much.

**Pros:**
1. The paper is concise and is clearly written.
2. Some details of PPO are discussed, which may require some thinking.
**Cons:**
1. The work suffers from a lack of novelty.
2. The investigated problem is trivial.
3. There is little analysis of why some settings are good compared with others.